# A Convolutional Neural Network for Electrical Fault Recognition in Active Magnetic Bearing Systems

**DOI:** 10.3390/s23167023

**Published:** 2023-08-08

**Authors:** Giovanni Donati, Michele Basso, Graziano A. Manduzio, Marco Mugnaini, Tommaso Pecorella, Chiara Camerota

**Affiliations:** 1Department of Information Engineering, University of Florence, 50139 Florence, Italy; michele.basso@unifi.it (M.B.); tommaso.pecorella@unifi.it (T.P.); chiara.camerota@unifi.it (C.C.); 2Department of Information Engineering, University of Pisa, 56122 Pisa, Italy; grazianoalfredo.manduzio@phd.unipi.it; 3Department of Information Engineering and Mathematics, University of Siena, 53100 Siena, Italy; marco.mugnaini@unisi.it

**Keywords:** active magnetic bearing (AMB), fault analysis, convolutional neural networks, fault dictionary, rotordynamics

## Abstract

Active magnetic bearings are complex mechatronic systems that consist of mechanical, electrical, and software parts, unlike classical rolling bearings. Given the complexity of this type of system, fault detection is a critical process. This paper presents a new and easy way to detect faults based on the use of a fault dictionary and machine learning. The dictionary was built starting from fault signatures consisting of images obtained from the signals available in the system. Subsequently, a convolutional neural network was trained to recognize such fault signature images. The objective of this study was to develop a fault dictionary and a classifier to recognize the most frequent soft electrical faults that affect position sensors and actuators. The proposed method permits, in a computationally convenient way that can be implemented in real time, the determination of which component has failed and what kind of failure has occurred. Therefore, this fault identification system allows determining which countermeasure to adopt in order to enhance the reliability of the system. The performance of this method was assessed by means of a case study concerning a real turbomachine supported by two active magnetic bearings for the oil and gas field. Seventeen fault classes were considered, and the neural network fault classifier reached an accuracy of 93% on the test dataset.

## 1. Introduction

Active magnetic bearings (AMBs) are being increasingly used across a broad range of rotodynamic applications, ranging from small turbomolecular pumps for medical applications to large compressors in the megawatt range for the oil and gas field. Compared with classic contact bearings, AMBs offer several significant advantages. AMBs achieve a significant reduction in friction and in the associated wear by levitating the rotor relative to the stator parts. The elimination of friction also enables higher rotation speeds and greater system efficiency and eliminates the need for cumbersome lubrication systems that are typically required for traditional bearings. Due to the inherently unstable nature of AMBs, a stabilizing feedback control is needed for proper functioning. Therefore, an AMB is a complex mechatronic system comprising various components, including the controller, position sensors, and actuators. The synthesis of the controller is a critical aspect of AMB systems. Various controller structures and design processes can be used, as reviewed, for example, in [1,2]. Among these, augmented PIDs are probably the most widespread choice in industry because of their versatility, accuracy, efficiency, and cost-effectiveness. All the controllers of AMB systems rely on precise measurements of the rotor position, for which a common choice is to use inductive or eddy current sensors, even if recently self-sensing sensors and optical sensors have also been introduced [3]. The effect of the choice of sensors is discussed, for example, in [4]. Regarding the other components, PWM is commonly used for driving the electromagnets that make the rotor levitate. The behavior of all the components in the loop contributes to determining the stiffness and damping of the bearings, which are directly related to the stability and performance of the closed-loop system. To improve the robustness of such a complex system, a fault detection and diagnostic system is advisable to ensure safe and reliable operation. As a result, several studies have developed methods to detect failures associated with the rotor or with the electrical and electronic components, as, for example, in [5,6]. Failure detection in AMB systems also enables the adoption of safety strategies exploiting their closed-loop architecture that can adapt to cope with the detected fault. In fact, due to their active nature, controllers can dynamically adjust the bearings’ behavior in real time. Exploiting the power of internal information processing, an AMB system can enhance its survival chances and reliability. For this reason, fault-tolerant AMB systems have been developed that can manage malfunctions, for example, using a reconfigurable control, as described in [6,7].

Usually, for fault diagnosis systems, the actual system behavior is compared with the expected one in nominal conditions to identify a faulty system condition. Observers of the system, as described in [5], are typically used.

Another common approach for identifying faults is the simulation-before-test technique. It involves constructing a fault dictionary from simulations of a particular plant, which collects examples of fault signatures that can be used to train a classifier capable of recognizing different faults, as described, for example, in [8]. 

Traditionally, fault diagnosis has been based on analyzing signals in the time and frequency domains. However, this paper proposes an approach that exploits a fault dictionary made by images of signals in the time domain to train a simple convolutional neural network that has the goal to recognize the AMB system’s faulty conditions. Taking the available electrical signals as sources, generalized orbits are built and converted into discrete 2D images that are used to fill the fault dictionary, with a technique that generalizes and extends the approach proposed by Xunshi, who used the sensors’ signals to build orbits to detect mechanical failures [9]. A similar method is proposed by Jing et al., who in their paper proposed a feature-based learning and fault diagnosis method for gearbox condition monitoring [10]. The fault features were obtained by using a simulation tool, developed by some of the authors [11], capable of automatically building the entire fault dictionary once the fault conditions are modeled. In the context of this work, only single electrical parametric faults have been considered for simplicity, but the fault classes can be extended to also include, e.g., mechanical faults, as in [12].

To exploit the knowledge stored in the fault dictionary, this paper proposes a classifier based on a convolutional neural network (CNN), well suited for image classification [13,14], trained with the fault signature examples. In the case of system fault, the trained convolutional neural network has the aim of identifying the faulty component and the type of fault that has occurred. 

Compared with other methods proposed in the literature that rely on analyzing signals in the time domain, the proposed approach, losing the time dependence, offers a novel solution to detect and locate faults, making full use of the potential of smart systems like AMBs. Moreover, with the help of computationally efficient image processing (Adam optimization) and simple neural networks, this automatic online diagnostic system can be developed without excessive computational cost. Such systems could be exploited in advanced prognostic maintenance systems to enhance the balance of plant capabilities over time, as described in [15,16,17]. 

An alternative approach is to employ 1D CNNs, which often exhibit superior performance, as evidenced in [18]. Nevertheless, for this specific study, a traditional CNN architecture was chosen. The primary objective was to develop a tool that enables image interpretation by both humans and neural networks.

Moreover, the decision to opt for the classic CNN network was influenced by the fact that 1D CNNs treat signals independently until the last layer, partially neglecting the crucial association that characterizes MIMO systems like AMB systems. By using a traditional CNN, these associations are taken into account, making the model more suitable for the specific task at hand. 

Another potential approach is the utilization of a Dislocated Time Series CNN (DTS-CNN), which has demonstrated superior performance in non-stationary conditions, as detailed in [19]. However, in the context of this study, only steady-state AMB systems are considered where nearly stationary conditions prevail. Since DTS-CNNs offer advantages in non-stationary settings, their benefits may not be fully realized in this steady-state context, making the classic CNN architecture a better choice for this study.

The paper structure is as follows: Section 2 describes the AMB modeling adopted to build a simulation tool. Section 3 presents the developed fault dictionary by which a classifier can be trained. Section 4 describes the proposed model of the image classifier. Section 5 presents a case study. Section 6 shows the case study results, and the conclusion follows. 

## 2. System Modeling

### 2.1. AMB Background

An AMB, applied to a turbomachine, has the purpose of making the rotor levitate with respect to the stator. Normally, a turbomachine is equipped with at least five AMB control axes: four for the rotor radial dynamics (two per radial bearing) and one for the axial one. Only the radial dynamics of the system have been considered since radial rotor dynamics are normally more complex than axial ones and for this reason have been chosen as the object of this study. An AMB is composed of two opposed electromagnets that attract a ferromagnetic object, in this case the rotor, and try to maintain it in the center of the air gap. A radial bearing is composed of two AMB control axes that are able to attract the rotor in any direction in a plane. Figure 1 shows a classical cross section of a radial AMB bearing in a heteropolar configuration, as described in [2]. 

The two opposed electromagnets are driven with a differential control current. Considering them, along the axis z, a linear relation can be found [20]: (1)$$F_{z}=k_{i}i_{z}-k_{s}Z$$
where $F_{z}$ is the magnetic force exerted on the rotor along the axis z; Z is the displacement along the same axis with respect to the nominal working conditions; and the parameters $k_{i}$ and $k_{s}$, respectively called electrical gain and negative stiffness, depend on the geometrical parameters of the bearings and on the nominal operating conditions, which are the nominal air gap $s_{0}$ and the bias current $i_{0}$. Equation (1) is more accurate the closer the working conditions are to the nominal ones. Because of the negative stiffness $k_{s}$, AMBs are inherently unstable. Hence, they are always inserted in a stabilizing closed-loop system. Figure 2 shows a block diagram of an AMB closed-loop system where the disturbances induced by each component are not included in the schematic. This system comprises amplifiers for driving the magnetic bearings, position sensors comprising the related conditioning electronics, and a controller. The position sensors constantly monitor the rotor position, while the controller utilizes these signals to calculate the necessary control signals for the actuators, which drive the magnetic bearings. The goal of the closed-loop system is to maintain the rotor in a levitated state at the center of the air gap by determining the appropriate currents $\bar{i}$ for each AMB, where $\bar{i}$ is the control currents’ vector.

### 2.2. AMB Closed-Loop System Modeling 

A state-space formulation was developed for each component to model the dynamics of the closed-loop system. The rotor radial state-space model was specifically constructed using a finite element method (FEM). This involved discretizing the rotor into N nodes along its geometric rotational axis. Timoshenko modeling was utilized for the shafts, with a detailed description of the formulation available in [21,22]. The resulting equation that described the rotor dynamics is the following:(2)$$M\ddot{\bar{q}}+\left(C+\Omega C_{g}\right)\dot{\bar{q}}+\left(K-K_{s}\right)\bar{q}={\bar{F}}_{AMB}+{\bar{F}}_{ext}\;{\bar{F}}_{AMB}=K_{i}\bar{i}$$
where M is the mass matrix, C is the damping matrix, $C_{g}$ is the gyroscopic matrix, K is the stiffness matrix, Ω is the rotor rotational speed, ${\bar{F}}_{ext}$ are the external forces acting on the rotor, ${\bar{F}}_{AMB}$ are the AMB control forces, $\bar{q}$ is the vector that represents the position of every node of the rotor, $\bar{i}$ the vector of the control currents, and $K_{i}$ and $K_{s}$ are, respectively, the matrix of current gains and the matrix of negative stiffnesses. Furthermore, a model order reduction technique was used on (2) to reduce the complexity of the system. In particular, modal truncation, described, for example, in [23], was used to eliminate all the rotor modes that were at a frequency above the range of interest for this type of application. Starting from (2), the rotor state-space equation can be determined:(3)$${\dot{\bar{X}}}_{R}=A_{R}{\bar{X}}_{R}+B_{R}\left({\bar{F}}_{AMB}+{\bar{F}}_{ext}\right)\;\bar{q}=C_{R}{\bar{X}}_{R}$$
where
$${\bar{X}}_{R}=\left[\begin{array}{c} \bar{q}\\ \dot{\bar{q}} \end{array}\right],\;\;A_{R}=\left[\begin{array}{cc} \left[0\right] & I\\ -M^{-1}\left(K-K_{s}\right) & -M^{-1}\left(C+{\Omega C}_{g}\right) \end{array}\right],\;B_{R}=\left[\begin{array}{c} \left[0\right]\\ M^{-1} \end{array}\right]\;,\;\;C_{R}=\left[\begin{array}{cc} I & \left[0\right] \end{array}\right].$$

Moreover, linear state-space models were used to describe the other components of the AMB system, considering their unique characteristics and specifications. Combining the sensors, controller, and actuators models, a second state-space system was generated:(4)$${\dot{\bar{X}}}_{B}=A_{B}{\bar{X}}_{B}+B_{B}\bar{q}\;{\bar{F}}_{AMB}=C_{B}{\bar{X}}_{B}$$
where ${\bar{X}}_{B}$ is the state, and ($A_{B}$, $B_{B}$, $C_{B}$) are the matrices of the second state-space model. Combining Equations (4) and (5), the state-space model of the whole closed-loop system was determined:(5)$$\frac{d}{dt}\left[\begin{array}{c} {\bar{X}}_{R}\\ {\bar{X}}_{B} \end{array}\right]=\left[\begin{array}{cc} A_{R} & B_{R}C_{B}\\ B_{B}C_{R} & A_{B} \end{array}\right]\left[\begin{array}{c} {\bar{X}}_{R}\\ {\bar{X}}_{B} \end{array}\right]+\left[\begin{array}{c} B_{R}\\ \left[0\right] \end{array}\right]{\bar{F}}_{ext}$$

The modeling of the whole system allows simulating the dynamic behavior of the whole system. The main contributions of the term ${\bar{F}}_{ext}$ are the unbalanced forces. Unbalanced forces are synchronous with the speed of the rotor and are induced by the presence of unbalanced masses with respect to the axis of rotation of the rotor. The unbalanced masses are due to inevitable manufacturing errors or rotor wear, and they are characteristic of a particular turbomachine. For a given axis x, at a fixed rotor speed Ω, the unbalanced force assumes the form:(6)$$F_{un_{x}}=\Omega^{2}Ucos(\Omega t+\phi )$$
where *ϕ* is the phase of the unbalance with respect to the other axis, and U is the unbalanced magnitude in kgm. In order to exactly reproduce the dynamic model, the machine was experimentally identified to accurately assess the imbalance and to fine-tune the state-space model (3). Some identification methods for AMB systems can be found in [24,25]. 

## 3. AMB Fault Modes

Due to the mechatronic structure of an active magnetic bearing (AMB) system, failures can manifest in various forms, including software, electrical, or mechanical. These faults can result in the high-speed rotor making contact with its housing, potentially compromising the safety of the entire plant. Although touch-down bearings are designed to prevent direct contact between the rotor and housing, such an event must be avoided at all costs. To address the diverse range of faults that can arise in AMB systems, several measures can be taken, including redundancies, quality control, individual measures, and various control strategies. Active fault diagnostics and corrections can also be employed. Guidelines for designing a reliable system are provided in ISO 14,839 [26] and API 617 [27] for turbomachinery on AMBs in the oil and gas field. A three-stage process for dealing with faults is described in [28], which involves determining the timing of the fault, identifying the faulty component, and identifying the type of fault. Given the complexity of the system, there are numerous reasons why malfunctions can occur, which can have different degrees of impact on system performance. In [29], the authors summarize the main causes of malfunctions, their occurrence, and severity. This study takes into consideration the most common electrical soft faults, which can be modeled in three different ways: multiplicative, bias, and noise, as discussed in [7,30]. The multiplicative type of error arises from the failure of system components, which results in changes in gains or sensitivities, commonly affecting amplifiers of conditioning electronics, sensors, power amplifiers, and actuators, usually due to temperature rise, fatigue, or short circuits [7]. Bias faults are related essentially to offset anomalous drifts related, e.g., to temperature rise, and noise faults are due to electrical and magnetic disturbances that are primarily introduced by the environment and predominantly impact the position sensors due to the low level of the transduced signals. More specifically, this study considered the most frequent scenario, i.e., that of a single fault at a time. Figure 3 provides a summary of the considered faults. 

## 4. Fault Dictionary

Steady-state signals, obtained from sensors and control systems, are exploited to build images that serve as fault signatures. In this study, the available considered signals were position signals from sensors, control signals from the controller, and current signals from the actuators, which are generally easily obtainable in an AMB system. Figure 4 summarizes the available electrical signals that were used to build the dictionary. Since all signals exhibit periodic behavior, during normal machine operations at a constant fixed rotational speed, it is possible to create groups of images by representing the signals related to one of the two orthogonal control axes as functions of those related to the other axis, disregarding time as an independent variable and obtaining generalized orbits of the signals related to each bearing, as described in Figure 4.

The fault features were obtained by using a simulation tool capable of automatically building the entire fault dictionary once the fault conditions were modeled. The simulation tool was implemented in the Matlab-Simulink environment and allowed us to perform Monte Carlo simulations by varying a selected set of component parameters using specific probability distributions or by adding noisy signals. The developed simulation tool was validated by comparing the results obtained with a commercial software (MADYN 2000 version 4.5).

Reference images, relative to non-faulty conditions, were formed by simulating machine operations at a constant rotor speed and collecting signals while varying the component parameters within their tolerance ranges and accounting for normal levels of noise. The dictionary was completed by simulating the different faults that belong to the previously introduced fault classes. In detail, examples of each single soft fault signatures in the fault classes listed above were obtained by randomly varying all the electrical parameters (gains, sensitivity, and biases) within the tolerance ranges, except for the faulty component which varied outside this range, or by injecting noise with an abnormal standard deviation.

The variations in the component parameters in the tolerance ranges determine regions of confidence in the reference images relative to non-faulty condition where the generalized orbits should remain under fault-free conditions. If the orbits go beyond these regions, a fault has occurred. The built fault dictionary consists of images containing the deformed orbits with respect to the reference ones, obtained by simulating the different faults for the particular plant. 

Once the simulations are completed, the feature images can be built. The time signals were normalized, and the orbit triplets were represented by images with a fixed resolution. Black-and-white images were generated through the Matplotlib library in Python, one for each orbit typology, one for the control signals, one for the position signals, and the last for current signals. At the end, these images were concatenated into a unique RGB image, so that each RGB channel concerns only one orbit typology. In other words, for every triplet of images, related to a particular scenario, a unique RGB color was associated at each orbit typology, the red for the position signals, the blue for control signals, and the green for current signals. Directly using images from available signals to train the classifier improves the ability of the system to understand the AMB system state, which offers a wider view and enhances the precision of the information captured.

## 5. Classification Algorithm: Convolutional Neural Network

The proposed classifier exploits a convolutional neural network (CNN), a neural network specialized for processing data that have a known, grid-like topology [31]. Examples include time series data, which can be thought of as a 1D grid, and image data, which are 2D grids of pixels. LeCun et al. [32] first applied the backpropagation algorithm to CNNs, with LeNet, a network developed to recognize handwritten digits. AlexNet [33] became the first modern deep convolutional neural network, representing a breakthrough in image classification.

A CNN is an architecture composed of multiple layers that collaborate to process and extract meaningful features from input data, each of them performing a different function in the network operations.

Convolutional layers consist of multiple filters, or kernels, that slide over the input data and perform convolutions. Each filter extracts specific features by detecting patterns and spatial information. The output of each filter is a feature map. An activation function, which introduces non-linearity into the network, is usually applied after each convolutional layer. Common choices include the Rectified Linear Unit (ReLU) or variants such as the Leaky ReLU or Parametric ReLU.

Pooling layers are frequently used after the convolutional layers to reduce the dimensionality of the feature maps and spatial dimensions. This down-sampling process helps to summarize and extract the most important information from the feature maps while reducing the computational complexity of the network. In other words, max pooling is a commonly used technique where the maximum value within a pooling window is selected as the representative value for that region.

After the convolutional and pooling layers, the feature maps are flattened into a one-dimensional vector. This vector is then fed into fully connected layers, also called dense layers, responsible for making predictions or classifications based on the extracted features. Like convolutional layers, activation functions are applied in fully connected layers to introduce non-linearity.

Finally, the last layer of CNNs is the output layer, which computes the network predictions. This time, the activation function used depends on the type of problem being solved. For classification tasks, as in this study, Softmax activation is commonly used to generate a probability distribution over the classes [34]. 

A loss function is used to measure the discrepancy between the predicted outputs and the ground truth labels. The choice of loss function depends on the problem type, such as categorical cross-entropy for multi-class classification, as in the proposed model [35].

During training, the parameters of the CNN are adjusted to minimize the loss function using optimization algorithms like stochastic gradient descent (SGD) or its variants, such as the Adam optimizer [36]. This process, known as backpropagation, updates the weights and biases of the network to improve its performance. The structure of a CNN, depicted in Figure 5, is very complex; however, many modern machine learning frameworks, like PyTorch, implement the above-described CNN operation.

### Proposed Model

In this study, the CNN classifier has the final objective of recognizing 17 fault classes, among which 16 are fault classes and 1 is the nominal class related to the normal operating conditions. These classes are summarized in Figure 6 and refer to Figure 3. 

The proposed CNN model consists of a series of alternately stacked convolutional, pooling, and fully connected layers, as shown in Figure 7. 

It employs convolutional layers to extract features, max pooling to down-sample the feature maps, and fully connected layers to map the extracted features to the output classes.

The first layer was a convolutional 2D layer, which has the images of orbit triplets as input channels, while the output channels were the 17 classes described in Table 1. In this way, the layer was able to expand the information of the provided data. With the same purpose, a second convolutional layer was implemented. Then, a max pooling was computed, halving the dimensions of the images both for considering the expanded information and for preventing the gradient explosion. At this point, another convolutional 2D layer increased the dimension again, from 16 to 36, followed by another max pooling, for the same reasons explained above. 

Subsequently, a batch normalization layer was applied to the extracted features in the previous layers, which allows the use of high learning rates and is quite unaffected by initialization. Moreover, this layer eliminates the need for dropout since it performs a regularization on the parameters, as explained in [37].

Afterwards, two additional convolutional layers were introduced. Unlike the preceding layers, these new layers aimed to combine or “zip” the output from the previous layer, thereby extracting the most significant features. This process helps to enhance the representation of the data by capturing more complex patterns and relationships.

The convolutional part of the network was concluded with a final max pooling operation. This step was followed by the transition to a fully connected layer. To prepare the data for this transition, the three-dimensional tensor resulting from the previous layers was flattened into a one-dimensional vector. Subsequently, three linear layers were trained in the network. Each of these layers was responsible for transforming the data and adjusting their dimensions to align with the output classes. In this case, the output dimension of each linear layer was rounded off to 17, which matches the number of classes in the classification task.

In the proposed model, each convolutional layer has a kernel dimension of 3 × 3 and a step dimension of 1 × 1, while the max pooling layers have a step dimension of 2 × 2. At least, each convolutional and fully connected layer, except the last one, was adhered with a ReLu activation function. The Adam optimizer was used, which is computationally efficient, has low memory requirements, is invariant to diagonal rescaling of gradients, and is well suited to problems with large data and/or parameters [36]. The architecture was implemented using the PyTorch libraries. 

## 6. Case Study

As a case study to assess the performance of the proposed diagnosis system, an AMB-supported system was considered. The case study involves a real, medium-size compressor supported by AMBs for the oil and gas field. Figure 8 illustrates the finite element model of the rotor under investigation with a mass of about 810 kg and a length of about 1.80 m. Regarding the electrical part, the controller structure was an augmented PID. It has a decentralized structure that divides the system into control axes, and every control axis is controlled independently. Regarding the other components, the system comprises switching pulse width modulation (PWM) amplifiers with heteropolar AMB actuators and inductive position sensors with a band of 4 kHz.

Under nominal conditions, the system features a constant rotor rotation speed of 7800 rpm and an imbalance of $2.10\times 10^{-3}kgm$ positioned at the center of mass of the rotor. The fault-free condition is given by variations in the sensor sensitivities within a 2% tolerance of the sensor, a bias below 1 μm, and in the presence of a Gaussian white noise with a standard deviation of 1 μm. As for the actuators, a 2% tolerance of the DC gain is taken into consideration. Figure 9 displays an example of nominal condition orbits related to the fault-free signatures for the two radial bearings of the rotor. 

The faulty conditions that were taken into account to form the classes of the fault dictionary were computed through Monte Carlo analysis using specific parameter distribution. Specifically, for each actuator multiplicative fault, the faulty actuator gain was chosen as a uniformly distributed random number out of the tolerance range centered in the nominal value in an interval of ±50% of the nominal value. Similarly, for each sensor multiplicative fault, the faulty sensor gain was chosen as a uniformly distributed random number out of the tolerance range centered in the nominal value in an interval of ±50% of the nominal value. For each sensor noise fault, the faulty sensor was subjected to a Gaussian additive noise with a standard deviation up to five times the nominal level of noise. Finally, for each sensor bias fault, the faulty sensor had a random bias amplitude larger than up to five times the nominal level of bias. Figure 10 shows an example of how orbits change shape when a specific fault occurs with respect to the orbits related to the nominal conditions reported, for example, in Figure 9. 

The dataset used in the case study consists of 37,600 RGB images obtained from different simulations, as described above in Section 3. More specifically, each image has a size of 3 × 224 × 224, where 3 represents the RGB channels and 224 × 224 is the number of pixels used for each image. Figure 11 gives an example of the RGB images used. The dataset is composed of 5600 examples of the fault-free condition and 2000 examples of all the other 16 fault classes. 

Before training, data were pre-processed with the aim of enhancing the model performance in terms of accuracy while also reducing noise interference. 

## 7. Results and Discussion

The dataset, obtained as described in the previous section, is composed of 5600 examples of the fault-free condition and 2000 examples of all the other 16 fault classes. It was divided into a training set, which includes 80% of the examples, and test and validation sets, each of which includes 10% of the remaining examples. 

The model was trained using the pre-processed training set for a total of 1000 epochs. The amount of time required for the training process was about 7 h using a GPU NVIDIA RTX A6000 48 GB GDDR6. Using the test set, the proposed CNN was validated; in fact, the test set accuracy reached was about 93%, while the training set accuracy reached was about 95%. These results are presented in Figure 12, with the test set accuracy depicted in orange and the training set performance in pink.

Figure 13 shows the test and accuracy ratio, which represents the ratio between the accuracy of the training and test reported in Figure 12. The descending trend observed in this plot suggests that the CNN’s estimated parameters are not affected by overfitting. This indicates that the model has achieved a good level of generalization, meaning it can effectively generalize its learned patterns and make accurate predictions on unseen data.

Additionally, in Figure 14, the obtained confusion matrix is reported, and Table 1 presents the codebook of the labels. The horizontal axis of the matrix represents the actual faults, while the vertical axis represents the predicted faults. The diagonal elements show the percentage of correct predictions; the values above the diagonal indicate false positives, while the lower ones represent false negatives.

As shown by Figure 14, the confusion matrix is quite diagonal even though the lowest values are about 70%. 

It was found that most of the misclassifications are related to soft faults related to small parametric deviations (with respect to the fault-free condition).

The achieved accuracy is quite satisfactory for soft fault recognition taking into account small parametric variations as well. This was due to two reasons. The first is that the regions of parameter variation related to faults border on the tolerance regions. The second reason is that the image resolution is limited. These facts led some classes not to be recognized correctly, specifically the faulty classes that are related to orbits that differ by a small amount with respect to the reference ones in the presence of a small variation in a parameter out of the tolerance ranges. This is a problem intrinsic to the definition of a soft fault itself. No net borders between the faulty and the fault-free conditions exist, but in any case, false positives or negatives related to these borderline situations have no severe consequences. 

## 8. Conclusions

Active magnetic bearings are gaining popularity in a range of rotating machinery applications due to their high performance and the elimination of cumbersome lubrication systems. On the other hand, to operate they need a complex closed-loop mechatronic system. So, to ensure safe and reliable operation, fault detection and diagnostic systems are necessary. Our proposed novel approach for fault diagnosing utilizes images of electrical signals available in an AMB system to train a classifier, a simple convolutional neural network, that was trained to detect the most common soft electrical faults. Remarkably, the proposed classifier exhibits high accuracy and generalizability without requiring an extensive amount of data. This approach can be easily extended to other fault typologies and to other AMB-supported systems. 

## Figures and Tables

**Figure 1 sensors-23-07023-f001:**
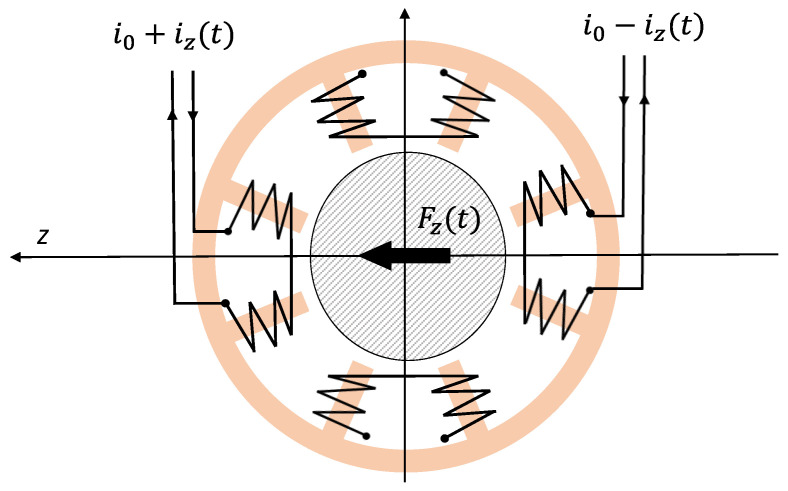
A classical cross section of a radial AMB in a heteropolar configuration.

**Figure 2 sensors-23-07023-f002:**
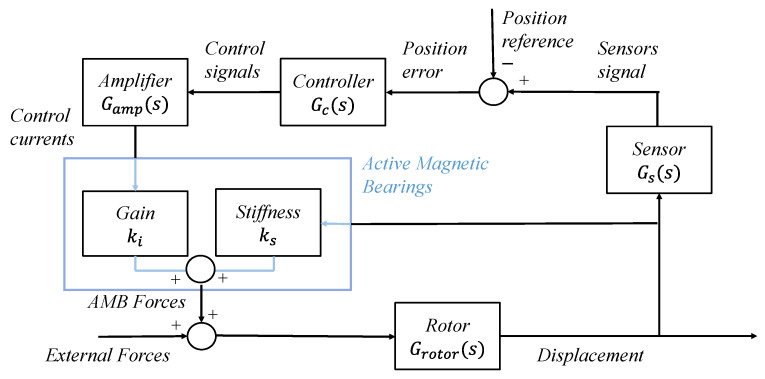
Block diagram of an AMB closed-loop system.

**Figure 3 sensors-23-07023-f003:**
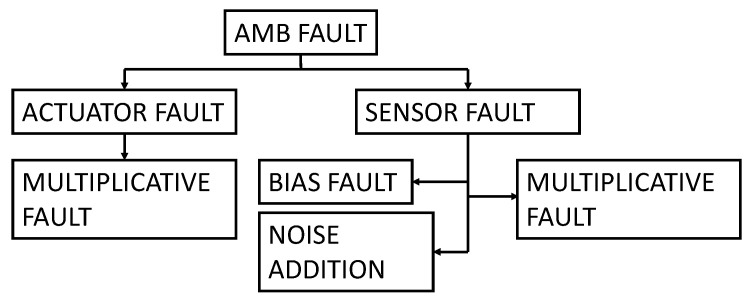
Summary of common electrical faults of AMB system.

**Figure 4 sensors-23-07023-f004:**
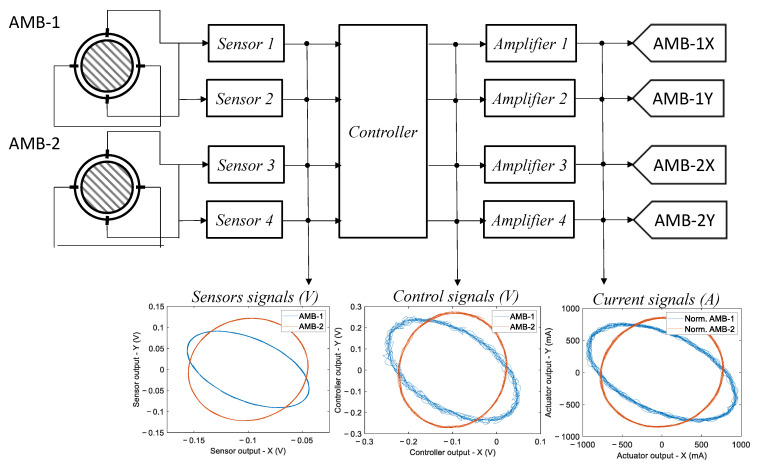
Summary of the available electrical signals used to build the dictionary.

**Figure 5 sensors-23-07023-f005:**
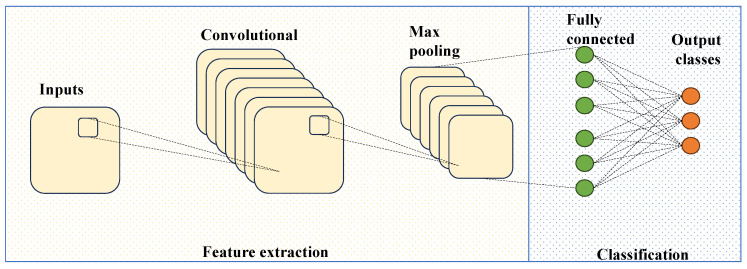
Schematic structure of CNN.

**Figure 6 sensors-23-07023-f006:**
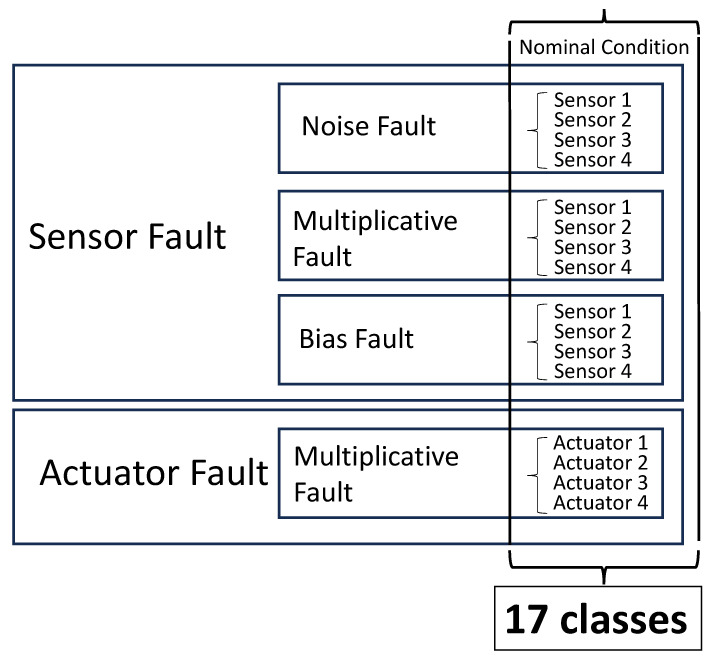
Summary of the classes to be classified.

**Figure 7 sensors-23-07023-f007:**
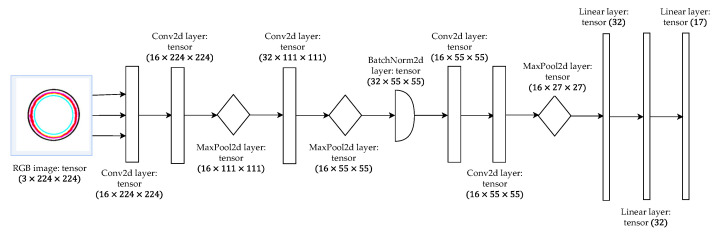
The proposed model structure.

**Figure 8 sensors-23-07023-f008:**
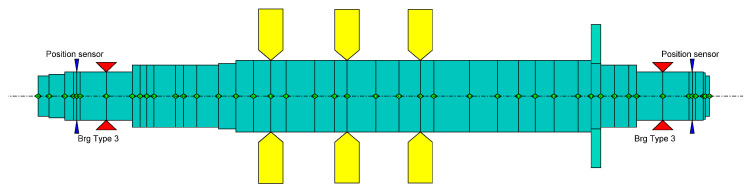
This image shows the finite element model of the rotor, where the red triangles are the bearings, the yellow elements are the disks, and the blue elements are the sensors.

**Figure 9 sensors-23-07023-f009:**
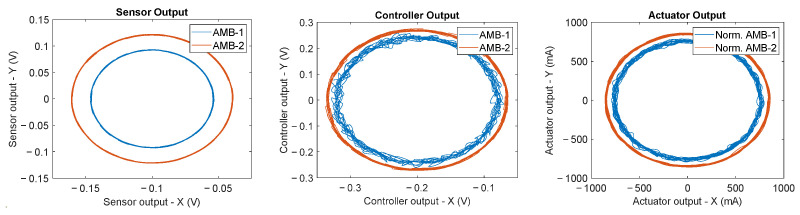
This image shows an example of nominal condition orbits. On the left is the sensor output, in the center the controller output, and on the right the actuator output.

**Figure 10 sensors-23-07023-f010:**
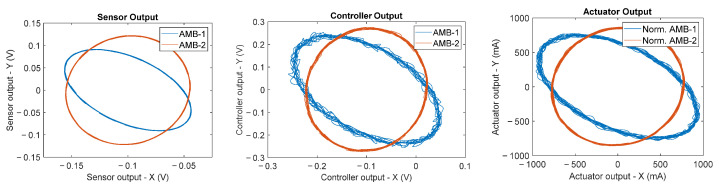
Example of signal orbits related to a sensor multiplicative fault, specifically an AMB-1 *x*-axis multiplicative fault of −50% with respect to the nominal value.

**Figure 11 sensors-23-07023-f011:**
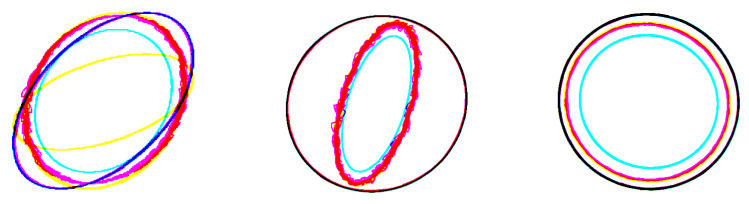
The RGB images used for training. In the right and central panel are shown, respectively, the actuator and sensor gain fault. In the panel on the left, the nominal condition is shown.

**Figure 12 sensors-23-07023-f012:**
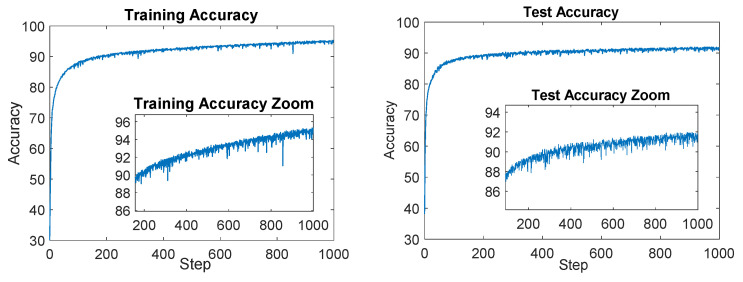
On the left the plot of the training accuracy for each epoch is shown; on the right the test accuracy for each epoch is reported.

**Figure 13 sensors-23-07023-f013:**
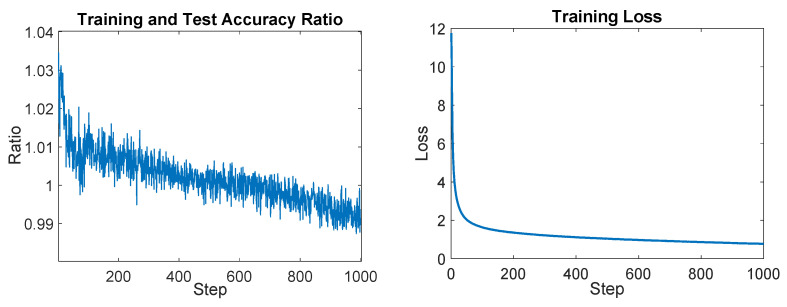
On the left the training and test accuracy ratio plot is shown, and on the right the training loss plot is reported.

**Figure 14 sensors-23-07023-f014:**
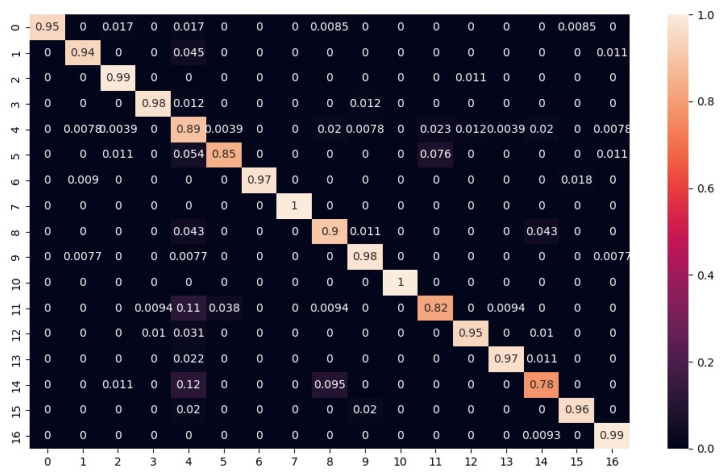
Confusion matrix of test set.

**Table 1 sensors-23-07023-t001:** Codebook of the confusion matrix.

ID	Component	Class	Class Size	ID	Component	Class	Class Size
0	Actuator 1	Gain Fault	2000	9	Sensor 2	Gain Fault	2000
1	Actuator 2	Gain Fault	2000	10	Sensor 2	Noise Fault	2000
2	Actuator 3	Gain Fault	2000	11	Sensor 3	Noise Fault	2000
3	Actuator 4	Gain Fault	2000	12	Sensor 3	Gain Fault	2000
4	-	Nominal Condition	5600	13	Sensor 3	Noise Fault	2000
5	Sensor 1	Bias Fault	2000	14	Sensor 4	Bias Fault	2000
6	Sensor 1	Gain Fault	2000	15	Sensor 4	Gain Fault	2000
7	Sensor 1	Noise Fault	2000	16	Sensor 4	Noise Fault	2000
8	Sensor 2	Bias Fault	2000				

## Data Availability

Data are available at the following link: https://drive.google.com/drive/folders/1Q-UdkmZOa6c0TgdSP9eNYTwq0Kc1lS9y (accessed on 2 August 2023).

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
