# Peer review of "A Convolutional Neural Network for Electrical Fault Recognition in Active Magnetic Bearing Systems"

_sensors, 2023, doi:10.3390/s23167023_

Round 1

Reviewer 1 Report

The topic is interesting and it has been presented with great attention to  quality and details, enlightning the original aspects of the work. I found the explanation excellent and the results convincing. Although the general topic of using CNN for fault diagnosis is not new, the application to the problem discussed in the paper seems in part original, even when compared to [9] in the paper.

Reviewer 2 Report

The suggested comments follow as,

1.  The research focus is sounds good. Suggested to highlight the clear objective of this research in abstract. 

2. Lacking to cover various types of Electrical Fault Recognition and important problems/issues identified in separate review section. 

3. Figure 2. Block diagram of an AMB closed loop system- Is this system induce any noise?

4. Figure 4. Summary of electrical signals that were chosen to build the dictionary-How the dictionary built?

5. Figure 6. Summary of the classes to be classified- Is these any specific reason for projecting 17 classes?

6. Input data and test bet not shown for Figure 12,13 and 14.  

Reviewer 3 Report

This paper presents a CNN based method to detect faults based on the use of a fault dictionary and Machine Learning. In particular, the dictionary was built starting from fault signatures consisting of images obtained from the signals available in the system. However, major corrections must be made to be considered in Sens:

-CNN has widely used for Electrical Fault Recognition. For instance, please check these related studies and discuss them in the introduction :

Junior, R.F.R., dos Santos Areias, I.A., Campos, M.M., Teixeira, C.E., da Silva, L.E.B. and Gomes, G.F., 2022. Fault detection and diagnosis in electric motors using 1d convolutional neural networks with multi-channel vibration signals. Measurement190, p.110759.

Liu, R., Meng, G., Yang, B., Sun, C. and Chen, X., 2016. Dislocated time series convolutional neural architecture: An intelligent fault diagnosis approach for electric machine. IEEE Transactions on Industrial Informatics13(3), pp.1310-1320.

- The training used 90% of the dataset. I afraid if the sizes of testing n validation are not enough. Is it possible to report the results of cross validation or at least 80/10/10?

- is the dataset available? Can you provide the link/reference.

- the authors used (the new method) to describe CNN. Please revise.

- the authors mentioned that the proposed method is efficient. Please discuss the computational cost of the proposed method compared to other/previous methods

- I would recommend to compare CNN results with other ML/DL methods/architectures to provide solid discussion. 

Minor editing is required 
